# Leveraging Latents for Efficient Thermography Classification and Segmentation

**Tamir Shor**[1]                  TAMIR.SHOR@CAMPUS.TECHNION.AC.IL

**Chaim Baskin**[1]                CHAIMBASKIN@CS.TECHNION.AC.IL

**Alex Bronstein**[1]                  BRON@CS.TECHNION.AC.IL

## Abstract

Breast cancer is a prominent health concern worldwide, currently being the second-most common and second-deadliest type of cancer in women. While current breast cancer diagnosis mainly relies on mammography imaging, in recent years the use of thermography for breast cancer imaging has been garnering growing popularity. Thermographic imaging relies on infrared cameras to capture body-emitted heat distributions. While these heat signatures have proven useful for computer-vision systems for accurate breast cancer segmentation and classification, prior work often relies on handcrafted feature engineering or complex architectures, potentially limiting the comparability and applicability of these methods. In this work, we present a novel algorithm for both breast cancer classification and segmentation. Rather than focusing efforts on manual feature and architecture engineering, our algorithm focuses on leveraging an informative, learned feature space, thus making our solution simpler to use and extend to other frameworks and downstream tasks, as well as more applicable to data-scarce settings. Our classification produces SOTA results, while we are the first work to produce segmentation regions studied in this paper. Code for reproducing all experiments is available at github.com/tamirshor7/Latents-Guided-Thermography.

**Keywords:** Thermography, Deep Learning, Medical Classification, Medical Segmentation.

## 1. Introduction

Many frameworks had been previously proposed to use thermography to perform breast cancer tumor classification, segmentation or both. (Lahane et al., 2021; Karthiga and Narasimhan, 2021; Abdel-Nasser et al., 2019) use a pre-chosen series of transformations and statistical/geometric feature extraction over the data to achieve downstream tasks. While efficient, these methods are relatively complex and difficult to implement. Furthermore, their use of hand-crafted feature extraction makes them difficult to generalize.

Other works leverage more powerful neural architectures for said task - (Mahoro and Akhloufi, 2024) employ Vision Transformers. (Alshehri and AlSaeed, 2022) use a CNN model interleaved with attention layers. Such methods are generally more difficult to train and require higher inference times compared to small CNN-backboned models. In the data-scarce setting of thermographic imaging, they impose additional difficulties as they usually demand more data to converge. Another line of works (UCUZAL et al., 2021; Yadav and Jadhav, 2022; Ornek and Ceylan, 2022) employs pre-trained CNN models followed by a small neural model to achieve the downstream task. This approach leverages a latent space trained on general computer-vision datasets, leaving target domain feature learning to the decoder, thus typically requiring correct choice of pre-trained encoder (or a subset of encoder layers), input resizing and normalization.

In this work we introduce a novel end-to-end algorithm for breast thermography. Unlike previous methods, our method removes the need of complex feature-extraction techniques or

costly training/inference resource demands, as it performs automatic, learned feature extraction, and leverages this feature space to quickly converge to downstream task performance. Our work makes the following contributions: Firstly, we perform tumor benign/malignant classification as to show our model achieves classification accuracy topping previous, more conceptually complex methods. Secondly, we show that a good latent space can be useful in few-shot settings, demonstrated on solving, for the best of our knowledge for the first time, a 7 region thermography-based segmentation.

Another key contribution of this paper is that unlike the vast majority of related prior work we encountered, we publish code to use our algorithm and reproduce all of our experiments.

## 2. Method

We use a simple yet resource and data-efficient encoder-decoder architecture elaborated in this section. A key consideration is the fact that labeled medical data, especially for thermograms, is scarce. We therefore design an unsupervised solution for our encoder, which is capable of leveraging large amounts of non-labeled data. Importantly, this encoder is decoupled from the decoder in training, making it applicable for various tasks and thermography datasets. As we later show, this feature space allows us to achieve our downstream task with only few-shot supervision, using only a lightweight UNet as our decoder.

### 2.1. CUTS Encoder

Our goal here is to use an encoder that performs efficient and potent feature extraction, however in a fully-learned manner and with lower implementation and resource complexity. To this end we leverage the recent CUTS encoder proposed by (Liu et al., 2022) - CUTS efficiently aggregates information from local and global regions for each pixel via a convolutional encoder, with contrastive learning guided training. This choice also allows encoding any type of thermal data representation.

### 2.2. UNet Decoder

Assuming we managed to create an informative latent space in 2.1, our goal here is to use a lightweight decoder to achieve any given downstream task in a supervised manner. We adopt the commonly used UNet (Ronneberger et al., 2015) architecture due to its speed and simplicity.

## 3. Experiments and Results

In our experiments we use the common DMR-IR thermography dataset published by (Silva et al., 2014). To test our method we train two different encoders - one with grayscale thermograms and one with RGB heatmap thermograms. Then, for each of the two considered tasks (classification and semantic segmentation) we train two decoders for each encoder - again one trained on each representation type. The reasoning behind this set of experiments is primarily to find the best setting for our method and show its applicability for both thermal representation types, however we also see importance in comparing both representation types for the encoding and decoding phases. This is because both types are widely used in literature, however to the best of our knowledge no works previously ablated usage of them. As results show, in our framework grayscale represetnations seem better for feature extraction, while heatmaps do better when decoded to achieve the downstream task.

| Encoded/Decoded Data Type | Classification | | | Segmentation | | | |
|---|---|---|---|---|---|---|---|
| | Accuracy | Precision | F1-Score | Accuracy | Precision | F1-Score | mIoU |
| G Enc. /w G Dec. | 0.978 | 0.989 | 0.987 | 0.878 | 0.851 | 0.791 | 0.662 |
| G Enc. /w H Dec. | 0.998 | 0.999 | 0.999 | 0.891 | 0.768 | 0.787 | 0.669 |
| H Enc. /w G Dec. | 0.927 | 0.997 | 0.957 | 0.882 | 0.787 | 0.791 | 0.665 |
| H Enc. /w H Dec. | 0.983 | 1.0 | 0.991 | 0.888 | 0.772 | 0.789 | 0.668 |

Table 1: **Classification and Segmentation Results** with different encoder/decoder data type combinations. $G$ means grayscale data was used and $H$ means heatmap data was used.

### 3.1. Classification

We use each encoder-decoder combination to classify benign/maligant tumors. Our classification model achieves 99.8% accuracy, surpassing the currently known SOTA of 99.7%, as published in a recent overview study by (Sayed et al., 2023). Nontheless, we stress the focus of this work is showing a flexible low-complexity model achieving performance at least on-par with SOTA. Full results are shown in table 1. Results indicate using grayscale data for encoding and heatmap data for decoding produces SOTA results, while reversing the roles produces worst results, stressing the importance of thermal data type choices.

### 3.2. Semantic Segmentation

We solve the task of segmenting 7 regions of interest - left/right breasts, left/right nipples, left/right armpits and neck. The motivation is assisting clinicians, or other models, in focusing on relevant per-pathology regions (e.g. breasts for tumor classification, armpits for cancerous lymphatic involvement). We show (table 1) our method produces accurate segmentation with only 52 labeled samples. We attribute this outcome to our potent latent space. Visual results for are portrayed in figure 1.

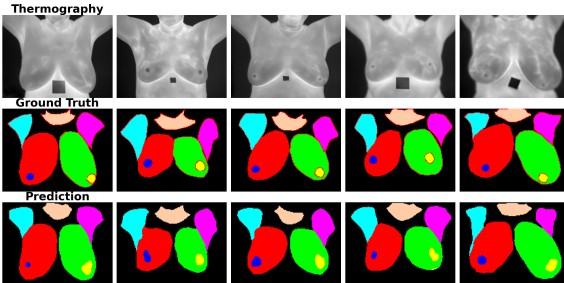

Figure 1: **Qualitative Segmentation Results** for best-performing model(grayscale encoder /w heatmap decoder), achieving segmentation similar to ground truth.

## 4. Conclusion

In this study we've performed thermography classification and segmentation to demonstrate how using a good feature space alongside a relatively simple decoder network can replace the need for more complex feature and architecture selection schemes. Furthermore, results show using grayscale data is generally better for feature extraction, while given a latent space, heatmaps are better for downstream tasks.

### Acknowledgments

This project has received funding from Horizon Europe - MISSIONS 80260. We also extend our gratitude to ThermoMind for supplying us with annotations used for the segmentation task.

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
