# OpenReview forum: "Leveraging Latents for Efficient Thermography Classification and Segmentation"
_MIDL.io/2024/Short_Papers — MIDL 2024 Short Papers_

### Official Review · Reviewer_dWX6 · 2024-04-24

**Confidence:** 5
**Final Rating:** 5

**Review:**

*** 01 Leveraging Latents for Efficient Thermography Classification and Segmentation
This submission proposes a novel algorithm for both breast cancer classification and segmentation using thermography. The authors argue that current methods rely on handcrafted feature engineering or complex architectures, making them difficult to implement and generalize. The work proposes a method that leverages a learned feature space, which is simpler to use and achieves state-of-the-art classification accuracy. The authors claim this is the first work performing segmentation of seven regions of interest in thermographic images.

The method achieves state-of-the-art classification accuracy and performs well on a segmentation task with few labeled samples. This suggests that the method is effective for both classification and segmentation tasks. The submission has a well motivated problem with adequate method, sufficiently described and convincing results. However, the paper, as is, suffers from overlength in the tables, breaking paper format and needs easy fixing. For these reasons, the recommendation is towards Acceptance.

---

### Decision · Program_Chairs · 2024-04-26

Accept